

# Effect of a HIIT protocol on the lower limb muscle power, ankle dorsiflexion and dynamic balance in a sedentary type 1 diabetes mellitus population: a pilot study

Jesús Alarcón-Gómez[1,*], Fernando Martin Rivera[1,2,*], Joaquin Madera[1] and Iván Chulvi-Medrano[1]

[1] Faculty of Physical Activity and Sports, University of Valencia, Valencia, Spain
[2] Research Group in Prevention and Health in Exercise and Sport, University of Valencia, Valencia, Spain
* These authors contributed equally to this work.

Corresponding author
Fernando Martin Rivera,
fernando.martin-rivera@uv.es

## ABSTRACT

**Background:** Type 1 diabetes mellitus (T1DM) is commonly associated with premature loss of muscle function, ankle dorsiflexion and dynamic balance. Those impairments, usually, lead to physical functionality deterioration. High-intensity interval training is an efficient and safety methodology since it prevents hypoglycemia and not requires much time, which are the main barriers for this population to practice exercise and increase physical conditioning. We hypothesized that a 6-week HIIT program performed on a cycle ergometer would increase lower limb muscle power, ankle dorsiflexion range of motion and dynamic balance without hypoglycemic situations.

**Methods:** A total of 19 diagnosed T1DM subjects were randomly assigned to HIIT group ($n = 11$; 6-week HIIT protocol) or Control group ($n = 8$; no treatment). Lower limb strength was evaluated through velocity execution in squat with three different overloads. Weight bearing lunge test (WBLT) was performed to test ankle dorsiflexion range of motion and Y-Balance test (YBT) was the test conducted to analyze dynamic balance performance.

**Results:** Velocity in squat improved a 11.3%, 9.4% and 10.1% ($p < 0.05$) with the 50%, 60% and 70% of their own body mass overload respectively, WBLT performance increased a 10.43% in the right limb and 15.45% in the left limb. YBT showed improvements in all directions (right limb-left limb): Anterior (4.3–6.1%), Posteromedial (1.8–5.2%) and Posterolateral (3.4–4.5%) in HIIT group ($p < 0.05$), unlike control group that did not experience any significant change in any of the variables ($p > 0.05$).

**Conclusion:** A 6-week HIIT program is safe and effective to improve execution velocity in squat movement, a fundamental skill in daily living activities, as well as ankle dorsiflexion range of motion and dynamic balance to reduce foot ulcers, risk falls and functional impairments. HIIT seems an efficient and safety training methodology not only for overcome T1DM barriers for exercising but also for improving functional capacities in T1DM people.

## INTRODUCTION

Prevalence of Type 1 Diabetes Mellitus (T1DM) is increasing worldwide (*You & Henneberg, 2016*). According to the International Diabetes Federation and World Health Organization, in the world, 25–45 million adults (>20 years old) suffered from T1DM. In reference to children and adolescents (0–20 years old), more than a million live with the disease, with 130.000 new diagnosed cases per year. It was estimated that the number of people with T1DM in the world will increase a 25% by 2030 (*International Diabetes Federation, 2019*; *WHO, 2016*). T1DM is a chronic metabolic disease characterized by the insufficient production of endogenous insulin and it is associated with multiple clinical manifestations that impair health (*Katsarou et al., 2017*). People with T1DM live in a continuous state of elevated glycemia (*Galassetti & Riddell, 2013*). This condition has been demonstrated to compromise skeletal muscle function, even beginning early in life in many cases, what would indicate that muscle dysfunction is a primary diabetic complication (being more accelerated with the development of neuropathy) (*Krause, Riddell & Hawke, 2011*; *Monaco, Gingrich & Hawke, 2019*). Hyperglycemia also promotes the increase in ankle stiffness even without the presence of neuropathy (*Francia et al., 2018*; *Searle et al., 2017*; *Searle, Spink & Chuter, 2018*) and compromises dynamic balance, also in young patients with no complications diagnosed (*Katsarou et al., 2017*; *Kukidome et al., 2017*; *Turcot et al., 2009*), among other physiological complications. Because of that, T1DM causes the premature loss of lower limb strength (*Celes et al., 2017*; *Monaco, Gingrich & Hawke, 2019*), functional capacities such dynamic balance (even in adults <50 years old) (*Camargo et al., 2015*; *D'Silva et al., 2016*; *Kukidome et al., 2017*) and joint mobility in comparison with their healthy counterparts (*Rao et al., 2006*). In T1DM population, low strength is related to functional limitations (*Lopez et al., 2018*; *Papa, Dong & Hassan, 2017*), ankle stiffness has been implicated as a potential factor of overload of the forefoot during the stance phase of the gait due to the reduced dorsiflexion capacity, which result in foot ulcers (*Rao et al., 2006*; *Searle et al., 2017*). Moreover, the increased postural sway caused by the pathology disorders increases the risk of falling (*D'Silva et al., 2016*; *Kukidome et al., 2017*). Thus, these physical consequences of T1DM could end in acute and chronic injuries and physical disability and frailty (*D'Silva et al., 2016*; *Vinik et al., 2017*; *Wu et al., 2020*).

Regular exercise is strongly recommended for people living with type 1 diabetes to prevent mainly comorbidities as the macrovascular (e.g. coronary arterial disease, peripheral arterial disease, and stroke) and microvascular complications (e.g. nephropathy, neuropathy, and retinopathy) of the disease, which lead to physical disability and premature death (*Farinha et al., 2017*; *Scott et al., 2019*). Nonetheless, the majority (>60%) of this population does not complete the general guidelines of exercise proposed by the American College of Sports Medicine (ACSM) and the American Diabetes Association (ADA) (*Leroux et al., 2014*; *Yardley et al., 2014*), which indicate at least 150 min of moderate to vigorous aerobic exercise per week and 2–3 non-consecutive sessions of

resistance training with a volume of 8–10 exercises, with 1–3 sets of 10–15 repetitions and an intensity of 50–75% of 1 repetition maximum (RM) (*Farinha et al., 2017*).

The most recurrent pretexts that T1DM people state for not exercising are the lack of time, the fear of a hypoglycemia event and loss of glycemic control due to inadequate knowledge about exercise variables management (*Lascar et al., 2014*). Those reasons make that few people with T1DM benefit from the improvement of aerobic capacity ($VO_{2max}$), insulin sensitivity, body composition, endothelial function, blood lipid profile, bone density and strength that aerobic and resistance exercises promote (*Codella, Terruzzi & Luzi, 2017*; *Scott et al., 2019*).

The aforementioned barriers that T1DM people face may be overcome with high-intensity interval training (HIIT), a training method that, despite being used since the early 20th century in sport performance, has been discovered to be an interesting tool for those with cardiometabolic diseases in the recent years (*Buchheit & Laursen, 2013*). HIIT involves repeated brief bouts of high intensity (>85% $VO_{2max}$) interspersed with passive or active recovery periods, requiring lower exercise duration than moderate-intensity continuous training (MICT), also HIIT prevents the drop of glycemia typical of MICT, due to its anaerobic predominance (*Farinha et al., 2017*). There is also evidence to suggest that HIIT elicits at least the same cardiometabolic effects in healthy and pathologic population that MICT does (*De Nardi et al., 2018*; *Hussain, Macaluso & Pearson, 2016*). These safe, effective and time-efficient results are sufficient to consider HIIT as an interesting form of training for the T1DM population. So far, HIIT has been tested in T1DM patients to analyze the long term effects in aerobic capacity and glycemic control (*Boff et al., 2019*; *Farinha et al., 2018*; *Scott et al., 2018*). However, this training strategy has not been investigated as a possible contributor of the development of strength, ankle dorsiflexion and dynamic balance in this population. It is known that HIIT is an interesting tool to improve the lower limb strength in older people (*Herbert et al., 2017*). Moreover, given that the ankle adopts a more dorsiflexed position when increase the intensity during cycling (*Holliday et al., 2019*) and dynamic balance is also enhanced with HIIT performed in cycle ergometer in older people (*Bellumori, Uygur & Knight, 2017*), the aim of this study was to investigate the effects of 6-week high-intensity interval training protocol on lower limb strength, ankle dorsiflexion range of motion and dynamic balance in an inactive T1DM population.

## MATERIALS AND METHODS

### Participants and research design

We recruited 19 inactive and clinically diagnosed as T1DM (10 males and 9 females) from the Valencian Diabetes Association (VDA) and social media announcement. Baseline characteristics of the sample are presented in Table 1. The following inclusion criteria were adopted: (1) aged 18–45 years, (2) duration of T1DM > 4 years, (3) HbA1C < 10% (4) no structured exercise training programs in the previous 6 months, (5) no known comorbidities not related to diabetes. In an a priori analysis of the required sample size (G*Power V.3.1.9.6), we needed 12 subjects per group, we had 11 participants in the
**Table 1 Lower limbs strength results.**

| Mean propulsive velocity | HIIT group ($n = 11$) | | Control group ($n = 8$) | | Size effect |
|---|---|---|---|---|---|
| | Pre | Post | Pre | Post | (C.I) |
| Body mass + 50% | 0.79 ± 0.17 | 0.87 ± 0.21* | 0.79 ± 0.14 | 0.80 ± 0.12 | 0.42 [−1.49 to 0.65] |
| Body mass + 60% | 0.74 ± 0.16 | 0.81 ± 0.17* | 0.76 ± 0.15 | 0.75 ± 0.14 | 0.49 [−1.62 to 0.64] |
| Body mass + 70% | 0.69 ± 0.16 | 0.76 ± 0.17* | 0.73 ± 0.19 | 0.71 ± 0.15 | 0.50 [−1.63 to 0.64] |

Notes:
* Statistical significance between pre-post within groups $p < 0.05$.
Data are presented in mean ± standard deviation m/s$^{-1}$.

experimental group and eight in the control group that is why we have titled the article as a pilot study.

Subjects excluded from the study include those who smoke regularly, take any medication that affects heart rate and those who had major surgery planned. Participants were informed of the purposes and risks involved in the study before giving their informed written consent to participate. Furthermore, they completed two questionnaires before the beginning of the measurement protocols: the PAR-Q to assess participants' level of risk to safely participate and the IPAQ (short version), to ensure the previous sedentary behavior of the subjects. The study procedures were in accordance with the principles of the Declaration of Helsinki and were approved by the Institutional Review Board of the University of Valencia which granted its approval to carry out the study within its facilities, IRB code: H1421157445503.

This is a randomized experimental, parallel design, open-label trial. The eligible subjects were randomly allocated (www.randomizer.org) to the experimental ($n = 11$, 38 ± 5.5 years, 5 men and 6 women, height 1.68 ± 0.09 m, body mass 70.5 ± 7.4 kg and 20.5 ± 8.4 years diagnosed) or control group ($n = 8$, 35 ± 8.2 years, 4 men and 4 women, height 1.69 ± 0.07 m, body mass 72.05 ± 5.0 kg and 21.1 ± 6.5 years diagnosed), and stratified/classified by gender to ensure a balanced number of men and women in each group. They were instructed not to change their nutritional habits and not to perform any regular exercise program outside of the study, which were not supervised. Initially, the control group had 10 participants but there were two drops out, a man and a woman, because of illness and pregnancy, respectively.

## Testing sessions

Firstly, all the participants performed an incremental test on a cycle ergometer (Excite Unity 3.0; Technogym S.p.A, Cesena, Italia) to determine peak power output (PPO) and peak oxygen consumption ($VO_{2peak}$) using a gas collection system (PNOE, Athens, Greece) that was calibrated in each test by means of ambient air. The PNOE system has proven its validity and reliability (*Tsekouras et al., 2019*). Before starting the test, capillary blood glucose concentrations were checked by their own blood glucose monitoring devices. They were told to arrive at the institutional gym with a glycemic level >100 mg/dl and less than 300 mg/dl in absence of ketones. If the glycemia was correct, the participant began the test normally. If not, the intake of 15–30 g of fast-acting

carbohydrates (CHO) we had available was compulsory when glycemia was <100 mg/dl and a small corrective insulin dose was used if hyperglycemia occurred without ketones. In the presence of ketones the exercise was canceled. Glycemia was checked again until the level of blood glucose was optimum to start the test. In the same way, it was recommended that patients not exercise at the peak of insulin action (*Scott et al., 2019*).

The test consisted in a warm-up of 5 min at 40 Watts (W). After that, the workload was increased by 20 W every minute until exhaustion. Participants were verbally encouraged to give their maximum effort during the exercises. The test ends with a cool down of 5 min at 40 W. Heart rate was continuously monitored by a Polar H10 (Polar Electro, Kempele, Finland). $VO_{2peak}$ was taken as the highest mean achieved within the last 15 s prior to exhaustion. Peak power output was registered to individualize the workloads in the experimental period training.

The hour of the day that each subject completed the test was recorded, as well as the menstrual phase of each female participant with the aim of repeating the same conditions in the second measurement to prevent their influence on the outcomes.

A total of 48 h after incremental testing, all the participants performed familiarization sessions to learn the correct technique to squat, as needed. This session consisted of the proper execution of the squat in a Smith machine and a reproduced experimental condition was taught, all monitored by the main researcher who is a strength and conditioning specialist. A week after the first test, the participants returned to the laboratory to perform three functional tests to measure lower limb strength, ankle dorsiflexion and dynamic balance: execution velocity performing squats with three different overloads; Weight Bearing Lunge Test (WBLT) and Y-Balance Test (YBT). The order of measurements was randomly counterbalanced to avoid any influence between them. Ten minutes of rest was set between tests to ensure the absence of all carry over effects.

Lower limb muscle power was measured by the execution velocity in the squat movement conducted in a Smith machine with no counterweight mechanism (Technogym S.p.A, Cesena, Italia) with overloads of 50%, 60% and 70% of the own body mass of each participant. The device selected to automatically calculate kinematic parameters of every repetition was an Encoder Speed4lift (Speed4lift S.L, Madrid, Spain). The encoder has proven its validity and reliability (*Pérez-Castilla et al., 2019*). After watching a standard video demonstration all the participants began a standard warm-up consisting of joint mobility, dynamic flexibility squats and lunges. After that, participants were required to perform three squats repetitions with each overload, conducting the concentric phase at maximal intended velocity and eccentric phase at controlled velocity. Subjects were instructed to perform parallel squats: descend until the inguinal crease was in projection with the top of the knee. Verbal and visual feedback was provided real-time to ensure the correct execution of the movement (*Pallarés et al., 2020*). The three sets were separated by 5-min rests to avoid the possible fatigue effect. Participants completed the test barefoot to avoid footwear influencing in the squat velocity. The data recorded and subsequently statistically analyzed were those corresponding to the highest execution

velocity of the three repetitions in the concentric phase of the squat of each overload measured.

The WBLT was performed to determine the ankle joint dorsiflexion range of motion. A tape line was placed on the floor perpendicular to the wall, where a vertical line was taped. Participants placed both hands on the wall in front of them and then aligned the center of the heel and the second toe of the foot that was being tested over the tape line. WBLT was conducted with the subjects barefoot to eliminate any influence of the footwear. Participants were asked to lunge forward trying to touch the vertical line on the wall with their knee without lifting the heel of the tested foot off the ground, but no encouragement was provided during the testing. Touching the vertical line perpendicular to the floor with the knee among the line formed by the heel and the second toe, helped to control subtalar joint movement and standardized the test between participants. The contralateral limb was positioned behind the testing limb in a comfortable position. The participants were allowed to perform three practice trials before the three test trials to achieve the farthest distance from the wall to the first toe, with 30-s rests between tests. The measurement was taken between the big toe and the wall, to the nearest 0.1 cm using a standard tape measure secured to the floor. The average of the three trials scores was documented as the test result. Both feet were measured (*Hall & Docherty, 2017*; *Langarika-Rocafort et al., 2017*; *Powden, Hoch & Hoch, 2015*; *Searle, Spink & Chuter, 2018*).

Finally, the YBT was conducted using a reliable standardized protocol. A "Y" was marked with tape on the lab floor to measure the dynamic balance in the Anterior (A), Posterolateral (PL) and Posteromedial (PM) directions. The posterior lines were marked 135° from A line, with 90° between them. Before testing, the participants performed 10 min of standardized warm-up, with 5 min of submaximal cycling followed by a dynamic stretch routine consisting of functional exercises: front-to back leg swing, side-to-side leg swing, lateral lunge, and sumo squat to stand (*Benis, Bonato & La Torre, 2016*), the participants were allowed to practice six times with each leg in each direction to minimize the learning effect. Afterwards, participants were asked to stand barefoot on one leg with the midfoot positioned over the central point and to reach with the contralateral leg as far as possible while hands were placed on the wing of the ilium. The reach distance was measured by marking the tape measure with erasable ink at the point where the most distant part of the foot reached. The trial was discarded and repeated if the subject (1) made a heavy touch, (2) rested the foot on the ground, (3) lost balance, or (4) failed to return to the starting position in a controlled manner. The process was repeated while standing on the other leg. The testing order was three trials standing on the right foot while reaching with the left foot in the A direction, followed by three trials standing on the left foot and reaching with the right foot in the A direction. The procedure was repeated for the PM and then the PL-reach directions. This order was proposed to avoid fatigue. The YBT scores were analyzed using the average of the last three trials for each reach direction for each lower extremity. Those values were normalized to the height of each participant which was measured with tallimeter (Seca™

709, Hamburg, Germany) (*Benis, Bonato & La Torre, 2016*; *Gribble & Hertel, 2003*; *Linek et al., 2017*; *Plisky et al., 2009*; *Shah et al., 2017*).

All tests were performed under similar environmental conditions (22 ± 1 °C, 40–60% humidity). The same test protocols were performed in exactly the same way after the experimental period by both control and HIIT groups.

## Training protocol

Training started the following week after completion of the pre-experimental procedures. Participants of the experimental group trained three times per week for 6 weeks under researcher supervision on a cycle ergometer (Excite Unity 3.0; Technogym S.p.A, Cesena, Italy). Heart rate while exercising was monitored with a Polar H10 (Polar Electro, Kempele, Finland) that was preconfigured with their heart rate zones. HIIT was a 1:2 protocol, which means that the high-intensity intervals lasted exactly half the time that the rest intervals did. The saddle height was always adjusted to the height of the subject's iliac crest. The training began with a 5-min warm-up at 50 W. Then, they performed repeated 30-s bouts of high-intensity cycling at a workload selected to elicit 85% of their individual PPO interspersed with 1 min of recovery at 40% PPO. The number of high-intensity intervals increased from twelve reps in weeks 1 and 2, to 16 reps in weeks 3 and 4, to 20 reps in weeks 5 and 6. Training ended with a 5-min cool down performed at 50 W. All sessions were supervised by, at least, a researcher. After the session, participants were told to check their glycemia level frequently and notify the investigators if a glycemia drop below 70 mg/dl occurred during the 24 h following the exercise (*Riddell et al., 2017*).

All sessions were supervised by the investigators and in order to reflect a real-world settings, researchers did not give advice about decreasing fast-acting insulin dosage or increasing carbohydrate consumption prior to each exercise session. Volunteers were only asked to arrive with glycemia <100 mg/dl (*Farinha et al., 2019*). Glucose levels were checked at least before and immediately after each exercise session, it was re-checked when glucose was not in the safety range. Fast-acting carbohydrates (15–30 g) were ingested when glycemia fell to ≤100 mg/dl. Hyperglycemia (250–300 mg/dl) was not set as a reason for postponing exercise if the patient felt well and ketones were negative (*Farinha et al., 2018*; *Scott et al., 2019*).

## Statistical analysis

All variables were expressed as a mean and standard deviation (M ± SD) and were analyzed using a statistical package (SPSS Inc., Chicago, IL, USA). Normality assumption by Shapiro–Wilks was identified for each variable. A mixed factorial ANOVA (2 × 2) was performed to assess the influence of "*condition*" (i.e., control group vs. experimental group) and "*time moment*" variable (i.e., pre-intervention, post-intervention) over Lower limb muscle power (LLS), Y-balance test (YBT), and Weight bearing lunge test (WBLT). In the event that Sphericity assumption was not met, freedom degrees were corrected using Greenhouse-Geisser estimation. Post Hoc analysis was corrected using Bonferroni adjustment. Hedges' G and the associated CI were used to assess the magnitude of mean

differences between control vs. experimental conditions. Significant differences were established at $p < 0.05$.

## RESULTS

The results of our study show the significant improvement, in all variables studied, obtained by the experimental group, while in the control group no change was observed between pre- and post-intervention conditions.

There were three mild hypoglycemia cases (67.9 ± 2.6 mg/dl) of 198 total trainings (1.5%), occurring immediately after exercise which only required a few minutes of rest and carbohydrate ingestion to be solved. No adverse cardiac events, respiratory events or musculoskeletal injuries were reported in the experimental period. There were no episodes of hyperglycemia, nocturnal hypoglycemia or episodes of diabetic ketoacidosis.

### Lower limbs muscle power

Lower limb Muscle Power experienced significant changes after the HIIT training period. With 50% of their body mass as additional load, the participants in the experimental group increased their speed of execution in squatting by 10.1%, the result of the mixed factorial ANOVA was $F(1,17) = 13.63$, $p = 0.02$. For 60% of the mass, by 9.4%, the result of the mixed factorial ANOVA was $F(1,17) = 13.56$, $p = 0.02$ and for 70%, there was an improvement of 10.1%, the result of the mixed factorial ANOVA was $F(1,17) = 20.21$, $p = 0.00$. In the control group, there were no significant changes between pre and post intervention assessment.

### Y-balance test

In the dynamic equilibrium, measured by the YBT, there were changes between both assessments in the experimental group (pre-post intervention). For the right leg, there was a significant increase ($p < 0.05$) of 4.2%, the result of the mixed factorial ANOVA was $F(1,17) = 7.1$, $p = 0.02$, and 3.4%, the result of the mixed factorial ANOVA was $F(1,17) = 4.73$, $p = 0.04$, in the anterior and posterolateral direction respectively. In contrast, in the posteromedial direction there was no significant change for this leg (1.9%). On the other hand, in the left leg, in the anterior direction there was an improvement of 6.1%, the result of the mixed factorial ANOVA was $F(1,17) = 13.6$, $p = 0.02$, in the posterolateral one there was an improvement of 4.5%, the result of the mixed factorial ANOVA was $F(1,17) = 9.1$, $p = 0.01$, and in the posteromedial one of 5.2%, the result of the mixed factorial ANOVA was $F(1,17) = 16.17$, $p = 0.01$. The control group experienced no change between the two assessments ($p > 0.05$). Data are presented in Table 2.

### Weight bearing lunge test

Ankle dorsiflexion improved by 15.4% in the left foot of the participants, the result of the mixed factorial ANOVA was $F(1,17) = 11.33$, $p = 0.04$, and by 11.3% in the right foot in the experimental group (pre-post intervention), the result of the mixed factorial ANOVA was $F(1,17) = 19.67$, $p = 0.00$. The control group experienced no change between the two assessments. Data are presented in Table 3.

**Table 2 Y-balance test results.**

| Y-balance test | HIIT group (n = 11) | | Control group (n = 8) | | Size effect |
|---|---|---|---|---|---|
| | Pre | Post | Pre | Post | |
| Anterior-right (cm) | 41.81 ± 2.28 | 43.61 ± 2.78* | 46.74 ± 5.56 | 47.20 ± 4.97 | 0.32 [−1.32 to 0.67] |
| Posterolateral-right (cm) | 47.08 ± 5.54 | 48.68 ± 5.08* | 45.91 ± 5.78 | 45.39 ± 3.35 | 0.36 [−1.38 to 0.66] |
| Posteromedial-right (cm) | 52.47 ± 4.82 | 53.49 ± 5.34 | 48.96 ± 6.71 | 48.93 ± 7.36 | 0.18 [−1.09 to 0.74] |
| Anterior-left (cm) | 44.94 ± 4.75 | 47.68 ± 4.87* | 50.52 ± 8.80 | 51.17 ± 6.34 | 0.30 [−1.28 to 0.68] |
| Posterolateral-left (cm) | 50.97 ± 5.54 | 53.30 ± 4.98* | 49.85 ± 4.70 | 50.29 ± 4.54 | 0.35 [−1.36 to 0.67] |
| Posteromedial-left (cm) | 49.20 ± 6.11 | 51.83 ± 5.45* | 46.43 ± 4.73 | 46.45 ± 3.76 | 0.45 [−1.54 to 0.65] |

Notes:
* Statistical significance between pre-post within groups $p < 0.05$.
Data are presented in mean ± standard deviation in centimeters (cm).

**Table 3 Weight bearing lung test results.**

| Weight bearing lung test | HIIT group (n = 11) | | Control group (n = 8) | | Size effect |
|---|---|---|---|---|---|
| | Pre | Post | Pre | Post | |
| **Right** (cm) | 11.5 ± 3.9 | 12.8 ± 3.4* | 11.0 ± 1.9 | 10.4 ± 2.2 | 0.56 [−1.76 to 0.64] |
| **Left** (cm) | 11.03 ± 3.3 | 12.7 ± 3.1* | 12.1 ± 1.5 | 11.6 ± 1.7 | −0.77 [−2.18 to 0.65] |

Notes:
* Statistical significance between pre-post within groups $p < 0.05$.
Data are presented in mean ± standard deviation in centimeters (cm).

# DISCUSSION

Our study demonstrates that a 6-week HIIT protocol is sufficient to result in functional capacity improvements in a previously inactive T1DM population without clinical impairments. Moreover, the study showed that this training method is safe for this population in field training since no insulin adjustments needed.

Only 3 of 198 total trainings, what means less than 1.5%, resulted in hypoglycemia, and they were mild cases (69.7 ± 2.6 mg/dl). These data suggest that HIIT prevents the blood glucose level dropped as well as previous studies reported and which is associated with catecholamine releasing and subsequent increase in hepatic glucose production which offsets the effect of hyperinsulinemia (*Boff et al., 2019*; *Farinha et al., 2017*, *2018*; *Scott et al., 2018*).

Impairments to skeletal muscle health in T1DM are, in many ways, similar to that observed in the muscle of aged individuals, but are occurring at a younger age in T1DM and compromises skeletal muscle function, sometimes since childhood (*Krause, Riddell & Hawke, 2011*; *Monaco, Gingrich & Hawke, 2019*). Our data is in agreement with previous research where reported the beneficial effects of applying HIIT in aging men was reported. Different studies by Peter Herbert and coworkers from different universities of the United Kingdom and Australia investigated the effect of HIIT performed on a cycle ergometer, as we did, on peak muscular power in both sedentary and active aging men. Given that similarity between older and T1DM muscle functionality, the results of those studies are in line with ours, since they observed that HIIT (6 × 30-s bouts at 90% heart rate

reserve interspersed with 3-min active recovery) performed every 5 days, for 6 weeks, induces a 8% increase in relative PPO measured with an incremental test in male masters athletes (*Herbert et al., 2017*). Other similar studies from this group, this time with sedentary seniors subjects analyzed the effect of the same type of HIIT aforementioned in muscle strength, resulting in an increase 14–17% approximately in relative peak power output, measured with an incremental test as well (*Sculthorpe, Herbert & Grace, 2015*, *2017*). Before the HIIT protocol, all the participants performed 6 weeks of pre-conditioning exercise protocol to prepare them for the high-intensity exercise. The mechanisms related to the lower limb muscle power gain after the HIIT protocol remain unclear, but we hypothesize that mechanic efficiency and neuromuscular capacity could be behind these adaptations (*Jabbour et al., 2017*; *Stöggl & Björklund, 2017*).

Despite the T1DM and age-related changes to muscle power manifests in an impairment of the functional fitness, no previous research has tested the hypothesis about the relationship between the HIIT performed in cycle ergometer and muscle power gains and improvement in functional movements in T1DM individuals.

To the best of our knowledge, this is the first study to investigate the impact of HIIT on lower limb muscle power in T1DM people. Our main result is that the subjects improved significantly their velocity in squat with the 50%, 60% and 70% of their own body mass overload respectively. Since squat movement pattern represents functional capacity in primary daily living tasks (*Myer et al., 2014*) such as sitting or standing up, our results show that people with T1DM improve those capacities through a 6-week HIIT protocol type 1:2 performed in cycle ergometer, which is accessible and safe for this population.

Additional findings of the present study show improvement in ankle dorsiflexion range of motion, confirming previously reported results of studies which observed that the ankle adopts a more dorsiflexed position with an increase in cycling intensity (*Bini & Diefenthaeler, 2010*; *Holliday et al., 2019*). In the current literature, there are no published investigations that analyze ankle dorsiflexion after a HIIT period in T1DM neither in healthy people either, so our results cannot be compared with previous ones. In this study, we found that a HIIT protocol performing 12–20 high-intensity intervals at 85% PPO and resting intervals at 40% PPO for eighteen sessions in 6 weeks was enough to improve ankle joint dorsiflexion in the right lower limb (10.4%) and in the left lower limb (15.4%) in T1DM people. We hypothesize that the difference in improvement between the left and right leg is due to the left being the non-dominant limb, with the exception of one woman, in the entire experimental sample, so that the initial performance in WBLT in that limb was worse than in the right one. This improvement is interesting because of a proper dorsiflexion range of motion is crucial to allow a correct functionality in daily living activities (*Medeiros & Martini, 2018*) and a very important factor in rehabilitation gait and for improved walking, particularly in clinical population (*Embrey et al., 2010*). It is also important to reduce the 12–25% risk that people with diabetes have to developing foot ulcers (*Searle et al., 2017*). It is possible to indicate that HIIT performed in cycle ergometer may be beneficial and an interesting tool for maintaining in non-impairment functionality in T1DM people.

Lastly, the results of this investigation have provided evidence that HIIT performed in cycle ergometer is an effective training strategy to improve dynamic balance in a T1DM population measured through YBT. Shorter reach distances performed in this test are typically associated with mechanical or sensorimotor system constraint (*Hoch, Staton & McKeon, 2011*) and reduced functionality to carry out daily living tasks (*Camargo et al., 2015*; *D'Silva et al., 2016*; *Shimada et al., 2003*; *Teyhen et al., 2014*). Data obtained with YBT reveals significant improvements in this test in both right and left limb. PM direction only was improved in the left limb (5.2%). Indeed, in A and PL directions important changes were reported in both limbs. Performance in those YBT directions are high correlated with ankle joint dorsiflexion (*Hoch, Staton & McKeon, 2011*; *Suryavanshi et al., 2015*), what is consistent with the results obtained in WBLT in which ankle dorsiflexion was increased by the participants. These results could show that people with T1DM without neuropathy can increase their dynamic balance end prevent future functional impairments and falls, given neuropathy development is not imperative to show reduced dynamic balance (*Abdul-Rahman et al., 2016*; *Kukidome et al., 2017*).

A possible explanation for the functional improvements could be, firstly, dorsiflexed position in cycling could be sufficient flexibility stimulus for reduced range of motion patients, and secondly for reducing glycation products in T1DM and in turn improving the joint health (*Abate et al., 2013*).

The practical implication of the present study is to bring inactive T1DM people closer to regular exercise. Our group has corroborated previous researches that reported HIIT as an effective training strategy to prevent hypo glycemia in T1DM people. Moreover, the functional improvements we have demonstrated using a 1:2 HIIT protocol indicate that it is a correct training prescription for T1DM people to start doing physical exercise. The results of this study demonstrate functional benefits of 6-week HIIT in cycle ergometer in T1DM subjects. Since there are no previous studies, our results should be taken with caution because this study has limitations that must be addressed. Limited sampled used with a specific range of age. We did not assess functional improvements in gait and finally we did no measure biomarkers related to the joint health in T1DM as the advanced glycation end-products (AGEs) that are actively produced and accumulated in the circulating blood and various tissues as tendons in chronic hyperglycemic situations as T1DM. As a contributing factor in T1DM complications its improvement could correlate with joint health. It also must be mentioned, that neuropathy increases the impairments in the capacities assessed in this work and it was not controlled given no participants were diagnosed with neuropathy. The control group did not perform any exercise during the experimental period, so their results were to be expected. Thus, it would be interesting to add a third group that performed continuous moderate exercise to analyze the difference with the HIIT group.

## CONCLUSIONS

In conclusion, 6-week HIIT protocol 1:2 type, performing high-intensity intervals at 85% PPO and active rest intervals at 40% PPO in a cycle ergometer for three sessions per week, apart from being accessible and safe since participants were able to complete all the

sessions with the intensity required without suffering any severe or undesirable episodes of hypoglycemia, was enough to improve lower limb muscle power, ankle joint dorsiflexion and dynamic balance in T1DM, which is related to an improvement in functionality in daily living activities. HIIT seems like an interesting approach for improving physical functionality in T1DM people.

### Funding
The authors received no funding for this work.

### Competing Interests
The authors declare that they have no competing interests.

### Author Contributions
- Jesús Alarcón-Gómez conceived and designed the experiments, performed the experiments, analyzed the data, prepared figures and/or tables, authored or reviewed drafts of the paper, and approved the final draft.
- Fernando Martin Rivera conceived and designed the experiments, performed the experiments, analyzed the data, prepared figures and/or tables, authored or reviewed drafts of the paper, and approved the final draft.
- Joaquin Madera performed the experiments, prepared figures and/or tables, authored or reviewed drafts of the paper, and approved the final draft.
- Iván Chulvi-Medrano performed the experiments, analyzed the data, prepared figures and/or tables, authored or reviewed drafts of the paper, and approved the final draft.

### Human Ethics
The following information was supplied relating to ethical approvals (i.e., approving body and any reference numbers):

The University of Valencia granted Ethical approval to carry out the study within its facilities (h1421157445503).

### Data Availability
The raw measurements are available in the Supplemental Files.

### Supplemental Information
Supplemental information for this article can be found online at http://dx.doi.org/10.7717/peerj.10510#supplemental-information.

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
