# Peer review of "Effect of a HIIT protocol on the lower limb muscle power, ankle dorsiflexion and dynamic balance in a sedentary type 1 diabetes mellitus population: a pilot study"

_PeerJ, doi:10.7717/peerj.10510_

## Round 0.1 · original submission · Major Revisions

I believe that this manuscript provides an important addition to the effects of high-intensity interval training (HIIT) on Type 1 Diabetes Mellitus patients. However, more information regarding how sample size was calculated, and additional statistical data (effects size, etc.) are needed to verify the statistical power. Whether the number of subjects should not be enough for an appropriate statistical approach I suggest adding in the title the term “a pilot study”.

Moreover, through the manuscript values should contain one significative digit (e.g. 14.3% and not 14.31%)

Reviewer 1 ·

Basic reporting

This manuscript aimed to investigate the effects of 6-week high-intensity interval training protocol on lower limb strength, ankle dorsiflexion range of motion and dynamic balance in an inactive T1DM population.
The present topic is new, interesting and relevant for the scientific knowledge, concretely for the readers of PeerJ, since the present results show that a 6-week HIIT program is safe and effective to improve execution velocity in squat movement which is a fundamental skill in daily living activities, as well as ankle dorsiflexion range of motion and dynamic balance to reduce foot ulcers, risk falls and functional impairments.
The authors used a correct English language but it needs to be slightly improved. There are some confusing words, and expression that avoid me to well understand specific answers. I consider that a revision performed by a native English speaker would increase the comprehension of many ideas.
I commend the authors for attaching the raw data.

Experimental design

The authors have met the manuscript structure following PeerJ author guidelines. The methods are correctly explained. Moreover, figures and tables are generally adequate and relevant.

Validity of the findings

The results are robust and support the main conclusion of the manuscript.

Additional comments

This manuscript aimed to investigate the effects of 6-week high-intensity interval training protocol on lower limb strength, ankle dorsiflexion range of motion and dynamic balance in an inactive T1DM population.
The present topic is new, interesting and relevant for the scientific knowledge, concretely for the readers of PeerJ, since the present results show that a 6-week HIIT program is safe and effective to improve execution velocity in squat movement which is a fundamental skill in daily living activities, as well as ankle dorsiflexion range of motion and dynamic balance to reduce foot ulcers, risk falls and functional impairments.
The authors used a correct English language but it needs to be slightly improved. There are some confusing words, and expression that avoid me to well understand specific answers. I consider that a revision performed by a native English speaker would increase the comprehension of many ideas.
The authors have met the manuscript structure following PeerJ author guidelines. Moreover, figures and tables are generally adequate and relevant. The results are robust and support the main conclusion of the manuscript.
Although the manuscript has a high scientific merit and accuracy, minor issues can be addressed to improve the quality and comprehensibility before its publication:
• Some statements in the introduction section should be supported by references (see lines 51 or 56 among others).
• Did the authors perform power analysis to determine sample size before the beginning of the study?
• The authors mentioned that the participants were instructed not to change their nutritional habits and not to perform any regular exercise program out of the study. However, were these variables experimentally controlled?
• Did the author control the physical activity levels of the participants before the exercise test? This fact could influence data collection.
• The authors reported that they measured VO2max but they have not reported data of cardiorespiratory fitness after the intervention. I suggest them to include these data.
• This reviewer recommend that the authors would include a paragraph discussing why a HIIT training program (which is usually applied to improve cardiorespiratory outcomes) is a valid tool to increase muscular strength. Which physiological mechanism are involved in this process?
• Finally, a specific paragraph including the practical implications of the study’ findings would be acknowledged.

·

Basic reporting

The reporting of the manuscript is understood, but there are some issues with the use of English language which would be assisted by having a person native to English language proof-read. Mostly the arguments are logical and appropriately referenced.

Within the discussion, there is some re-stating of results. It would be better to summarise the result in the discussion rather than directly re-stating it. There is no need to present data or p values in the discussion.

While the tables are appropriate, they could easily be combined into a single table.

Some more baseline data about the participants are required to better understand the context of the results and assist with interpretation. For example, duration of diagnosis, medications that participants were taking, exercise/physical activity being completed, etc. VO2peak is reported to be measured but has not been reported, it should be.

Experimental design

An appropriate experimental design was implemented, but it is unclear how the sample size was determined. Some reporting on sample size calculation should be included.

The methods are appropriately described but would benefit by some reporting of the variability of measurement, particularly when multiple measures were conducted (e.g. coefficient of variation).

To ensure full reporting and interpretation of the data, it would be useful to include some of the training data to verify that participants complied with the exercise prescription.

The authors claim to measure muscle strength, however the assessments conducted are probably more reflective of muscle power. Suggest changing all reference of muscle strength to muscle power.

Validity of the findings

The findings appear to be valid and mostly based on reported data. Claims of safety of the exercise intervention need to be better supported by well defined data presented in the results.
The discussion is appropriate and proposes more questions and hypotheses. However, the conclusion contains too much speculation and/or unsupported claims. Adding some of the identified data would partially mitigate this.

Additional comments

Line 55 – “endogen” should be ‘endogenous’
Line 69 – which counterparts?
Line 69/70 – Which population?
Line 79 – “dead” should be ‘death’
Line 95 – is “XX” meant to be 20th or something different?
Lines 343-345 - if this is an important finding, there needs to be something reported in the results about this. The remainder of this paragraph is reported more like results than discussion.
Lines 369-370 – better to summarise the result rather than re-state.

---

## Round 0.2 · Minor Revisions

The manuscript was improved. However, there are still some minor issues that should be resolved.

Reviewer 1 ·

Basic reporting

The authors have correctly answered all my concerns.

Experimental design

The authors have correctly answered all my concerns.

Validity of the findings

The authors have correctly answered all my concerns.

Additional comments

The authors have correctly answered all my concerns.

·

Basic reporting

The writing is much better following an English language review, however there are still some minor issues. Throughout the manuscripts, there are still regular references to muscle strength as an outcome of this study, when the authors have really measured muscle power outcomes rather than strength. Please read through the manuscript carefully and change.

Experimental design

From the authors response, I understand that as sample size of 11 was predicted to be required. This information has not been included in the manuscript but it should be. The authors should also identify what data they used to estimate the sample size.

Validity of the findings

Information pertaining to adverse events and safety has now been included to support statements, but these findings should be outlined in the results, before being explored in the discussion, not just simply reported in the discussion.

This prompted a closer look at the statistical analyses and results. The authors have indicated that they performed a mixed factorial ANOVA to assess the influence of condition and time, but the results have not been presented that way. From the results, it appears that separate pre-post (potentially t-tests) have been conducted for each group, so it is not clear if there was or was not any influence of condition. The authors should clarify the reporting of results. Particularly as the effect sizes and their (assume 95%) confidence intervals tend to indicate potentially trivial findings.

---

## Round 0.3 · accepted · Accept

Manuscript was improved and all the reviewers comments have been addressed

·

Basic reporting

The authors have appropriately addressed all of the reviewers comments and the manuscript is acceptable for publication in my opinion.

Experimental design

No additional comment

Validity of the findings

No additional comment